# Determining the Optimal Vacuum Frying Conditions for Silver Herring (*Spratelloides gracilis*) Using the Response Surface Methodology

**DOI:** 10.3390/foods12193533

**Published:** 2023-09-22

**Authors:** Hung-I Chien, Chiu-Chu Hwang, Yi-Chen Lee, Chun-Yung Huang, Shu-Chuan Chen, Chia-Hung Kuo, Yung-Hsiang Tsai

**Affiliations:** Department of Seafood Science, National Kaohsiung University of Science and Technology, Kaohsiung 811213, Taiwan; aaoc1018@yahoo.com.tw (H.-I.C.); omics1@gmail.com (C.-C.H.); lionlee@nkust.edu.tw (Y.-C.L.); cyhuang@nkust.edu.tw (C.-Y.H.); litengren273@gmail.com (S.-C.C.); kuoch@nkust.edu.tw (C.-H.K.)

**Keywords:** vacuum frying, response surface methodology, oil content, colour, fracturability

## Abstract

Vacuum frying (VF) is a selective technique for producing high-quality fried food that is mostly used on vegetables, fruits, and potato chips. It is rarely applied to the production of aquatic (especially fish) products. The purpose of this study is to explore whether VF technology can be applied to the preparation of dried silver herring products and to obtain the optimal VF conditions. The response surface methodology (RSM) was used to examine the factors affecting the quality of silver herring (*Spratelloides gracilis*) products after VF, namely temperature (75, 90, and 105 °C), duration (25, 35, and 45 min), and concentration (0, 15, and 30%) of maltose solution used to immerse the samples during pre-processing. The results indicated that VF temperatures had significant impacts on water activity (Aw), moisture content, yield, oil content, lightness (*L** value), and colour difference (*ÄE*). The higher the VF temperature, the lower the Aw, moisture content, yield, and oil content of the product, but the higher the *L** value and Δ*E*. Next, a longer VF duration resulted in higher oil content of the product. Maltose concentration was significantly and positively correlated with the yield and fracturability of the product. RSM analysis indicated that the optimal combination of processing conditions was a VF temperature of 105 °C, VF duration of 25 min, and maltose concentration of 27%. Under these VF conditions, the silver herring products had a moisture content of 3.91%, Aw of 0.198, oil content of 21.69%, *L** value of 28.19, Δ*E* of 27.31, and fracturability of 359 g/s. In summary, vacuum frying technology is suitable for the preparation of dried silver herring products, and this study can provide the optimal processing conditions for seafood processors to obtain better quality.

## 1. Introduction

The silver herring (*Spratelloides gracilis*) is a small pelagic and migratory fish that is widely distributed in the Indo-West Pacific Ocean, and can be caught in the offshore areas surrounding Taiwan [1]. It is a fast-growing and abundant fish species. Being small in size (average body length of approximately 5–6 cm) and rich in calcium, the entire fish is usually sold for consumption in a fresh, dried, or salted state [1]. In general, the traditional method of making dried silver herring in Taiwan is to pour the fresh fish into boiling 10% NaCl water and cook it for 3 min, then expose the fish body to the sun for 8–12 h until dry, or dry the fish with hot air at 50–60 °C for 3–6 h [1]. The dried fish then serves as a snack food or cooking ingredient [1]. 

Traditional frying (TF), as a drying method, involves heating oil under normal pressure to 160–180 °C before the moisture in the ingredients is dried by boiling and vaporising upon coming into contact with air [2]. The physicochemical qualities of the food change during the frying process, causing the products to have shortcomings such as a rancid odour, dark brown colouration, and shrunken shape [3]. In addition, the quality of frying oil deteriorates with excessive heating, which adversely affects the flavour, stability, colour, and texture of the food. Such products may also be harmful to human health [4,5,6].

Vacuum frying (VF) is another method of frying and drying. VF is different from TF in that the frying and heating processes occur inside a decompressed and airtight container [3]. Because the boiling point of water decreases when in a decompressed state, moisture is rapidly vaporised and removed in the vacuum system even with lower oil temperatures. Consequently, the food is dehydrated and dried to become crisp, while the original colour and flavour are maintained [7,8,9]. Moreover, decompression reduces the boiling point of the oil (that is, a drop in the temperature required for frying), which effectively prevents its oxidation and deterioration, thereby prolonging the service life of the frying oil and saving money [8,9]. Currently, common VF products include chips made from fruits, mushrooms, sweet potatoes, yams, carrots, and potatoes [4,8,10].

Generally, food ingredients undergo pre-processing prior to VF for the purpose of promoting dehydration, reducing oil content, maintaining colour, and increasing crispness [3,4]. Common pre-processing methods before VF include sugaring, salting, freezing, and blanching. Sugaring refers to immersing the ingredients in a sugar solution (maltose is commonly used) before VF. The sugar molecules enter the ingredients owing to osmotic pressure, whereas water molecules flow out to cause dehydration. This process reduces oil absorption during subsequent frying [4]. By reducing the oil content of VF products, sugaring helps to better maintain their colour and crispness [4,10]. The principle of salting is similar to that of sugaring, except that it will make the products taste salty. It is generally less used in the production of vegetable, fruit, and starch (potato) chips. Freezing converts the moisture content within the tissues of food ingredients into ice crystals. These sublimate and accelerate dehydration when the temperature rises and the pressure drops during frying. Concurrently, the tissues become hollow and porous, thereby increasing the crispness of the product [3,4,10]. During blanching, the ingredients are heated with boiling water for at least 30 s, which fixes their shape and maintains the colour and appearance of the finished product. For example, green vegetables must first be blanched before VF to better maintain their green lustre [3,4,10].

The response surface methodology (RSM) is used to examine the impact of various factors on the response during a multi-factor experiment. It involves appropriate experimental design, mathematical inference, and systematic analysis, and aims to determine the lowest number of experimental groups to obtain the optimal processing conditions [11,12]. Specifically, RSM predicts the corresponding results produced under different control factor settings. It does so by establishing the functional relationship between the factors and response, thereby arriving at the optimal response or operating conditions [11,12].

Most of the foods used for VF research are starches (such as potatoes) or fruits and vegetables (such as carrots) [13,14,15]. However, VF research using fish as the ingredient are rare [16,17]. The purpose of this study is to explore whether VF technology can be applied to the preparation of dried silver herring products and to obtain the optimal VF conditions. RSM was used to design experimental conditions involving three factors and three levels per factor, namely (i) VF temperature (75, 90, and 105 °C), (ii) VF duration (25, 35, and 45 min), and (iii) maltose concentration used for pre-processing (0, 15, and 30%). The selection of the above factors and levels is mainly based on our previous preliminary experimental results in the laboratory and the research results of Yang (2004) [18]. The aim was to examine the impacts of these factors and levels on the quality of the finished products. RSM calculation and analysis were also used to arrive at the optimal processing conditions.

## 2. Materials and Methods

### 2.1. Preparation of Silver Herring Samples

Fresh silver herring were purchased from a supermarket in Kaohsiung City, Taiwan, placed in crushed ice, and transported back to the National Kaohsiung University of Science and Technology laboratory. Upon arrival, the fish were immediately boiled in 90 °C salt water (1.0% NaCl) for 30 s, according to Yang (2004) [18]. After removal, the fish were drained dry, cooled to room temperature, and then stored in a freezer for subsequent use. Palm oil was used for frying the samples in the experiments.

### 2.2. VF Experiments

The vacuum fryer used in the study was a Type VF-5 (Chin Ying Fa Co., Ltd., Changhua, Taiwan). The VF level was set to 6.65 kPa, the centrifugal de-oiling speed was 1000 rpm, and the de-oiling time was 1 min. RSM using a Box–Behnken design (BBD) (a factorial design) was used to investigate the response patterns and also to determine the optimum synergy of variables. The BBD model of RSM was applied to establish a tri-factor, tri-level design, namely VF temperature (X_1_; 75, 90, and 105 °C), VF duration (X_2_; 25, 35, and 45 min), and maltose concentration during pre-processing (X_3_; 0, 15, and 30%). A total of 15 sets of experiments were designed (Table 1). The selection of the above factors and levels is mainly based on our previous preliminary experimental results in the laboratory and the research results of Yang (2004) [18]. Three replicate experiments were conducted in each treatment group.

For the sugaring step, the boiled fish samples were immersed in maltose solutions of various concentrations at 4 °C for 1 h. The impacts of the three processing factors of VF, namely temperature (X_1_), VF duration (X_2_), and maltose concentration (X_3_), on the quality of the fried silver herring products were then examined. The use of these three processing factors is based on Yang’s (2004) research [18]. The responses for sample testing included water activity (Aw), moisture content (%), yield (%), oil content (%), lightness (*L** value), colour difference (Δ*E*), and fracturability (g/s).

### 2.3. Quality Analysis of the Fried Products

#### 2.3.1. Water Activity (Aw)

After 10 g of the sample was finely ground, 1.0 g of the finely ground sample was placed into a plastic container for Aw determination using the AquaLab Model Series 3 (Decagon, Pullman, WA, USA) water activity meter [19]. The Aw measurement of each treatment sample was performed in duplicate.

#### 2.3.2. Moisture Content

Approximately 5 g of the finely ground sample was placed in an aluminium dish, and the moisture content was determined using an infrared moisture analyser (Model HE53, Mettler Toledo GmbH, Greifensee, Switzerland). The moisture content of the sample was recorded when a constant weight was reached [20]. The moisture content measurement of each treatment sample was performed in duplicate.

#### 2.3.3. Yield

The yield is the ratio of the weight of the sample after frying to its weight before frying and is calculated as a percentage (%). The weight of the sample was measured with an electronic balance (Mettler Toledo GmbH, Greifensee, Switzerland) [21]. The yield measurement of each treatment sample was performed in duplicate.

#### 2.3.4. Oil Content

Oil content was extracted from the sample with petroleum ether using Soxhlet extraction equipment (Soxtec System 2055 Tecator, FOSS, Hillerod, Denmark) and then gravimetrically determined [21]. The oil content measurement of each treatment sample was performed in duplicate.

#### 2.3.5. Colour Analysis

A colour meter (CR-300 Chroma meter, Konica Minolta, Inc., Tokyo, Japan) was used to measure the colour values of the sample. The indicators included lightness (*L**), +red to −green (*a**), and +yellow to −blue (*b**). Δ*E* was calculated based on the *L**, *a**, and *b** values using the following equation [21]:(1)$$\Delta E=\sqrt{{\left(L^{*}-L_{0}^{*}\right)}^{2}}+{\left(a^{*}-a_{0}^{*}\right)}^{2}+{\left(b^{*}-b_{0}^{*}\right)}^{2}$$
where *L*_0_*, *a*_0_*, and *b*_0_* refer to the colour values of the sample before VF; and *L**, *a**, and *b** are the colour values after VF. The colour measurement of each treatment sample was performed in duplicate.

#### 2.3.6. Texture Measurement

The texture profile analysis of the samples was measured with a Brookfield CT3 texture analyser [16]. The measurement item was fracturability (g/s), with the conditions set as follows: (i) probe: TA 39 cylinder, 2 mm D, 20 mm L; (ii) target value: 4.00 mm; (iii) predicted speed: 2 mm/s; (iv) test and return speeds: 1.5 mm/s; and (v) trigger point load: 2 g. In this study, the parameters and conditions used for texture analysis are mainly modified from the research of Su et al. [21]. The fracturability measurement of each treatment sample was performed in duplicate.

### 2.4. Optimization Procedure

Optimization of processing conditions is based on responses of moisture content, oil content, texture (fracturability), and *L** value. The numerical optimization techniques are used to optimize multiple responses simultaneously. The desired goal for each processing parameter and response is chosen, and all processing parameters are maintained within the specified parameter range. The target values on oil content and moisture content were minimized, while target values on *L** value and fracturability were maximized. The response surface plot was prepared after the optimal regression model was obtained using the numerical optimization tool of Design-Expert 7.0 software.

### 2.5. Statistical Analysis

The Design-Expert 7.0 (Stat-Ease Inc., Minneapolis, MN, USA) statistical software was used to analyse the results of the various factor combinations and determine the optimal conditions. Data obtained from sample analysis were using an analysis of variance (ANOVA) to study the effect of processing conditions on the responses of vacuum fried silver herring. The probability level *p* < 0.05 indicated that the difference was significant. In addition, a regression analysis was performed on the experimental data and expressed using an empirical second-order polynomial model, as follows:(2)$$Y=\beta_{0}+\sum_{i=1}^{3}\beta_{i}X_{i}+\sum_{i=1}^{3}\beta_{ii}{X_{i}}^{2}+\sum_{i=1}^{2}\sum_{j=i+1}^{3}\beta_{i}{}_{j}X_{i}X_{j}$$
where *Y* is the predicted response, ***β***_0_ is a constant, *β_i_* is the linear coefficient, *β_ii_* is the quadratic coefficient, *β_ij_* is the interaction coefficient of variables *i* and *j*, and *X_i_* and *X_j_* are independent variables.

RSM was used to analyse the impacts of VF frying on the samples’ parameters, including Aw, moisture content, yield, oil content, *L** value, Δ*E*, and fracturability. BBD was used to analyse the response (*Y*) of these qualities and obtain the optimal VF conditions. Long et al., (2014) [22] concluded that the larger the F value of the model and the smaller the P value, the more significant the impact of this factor on the response (*Y*). Jentzer et al., (2015) [23] stated that the coefficient of determination (*R*^2^) represents the goodness-of-fit that the prediction model of the regression method has on the experimental data. The closer the value of *R*^2^ to 1, the greater the reliability of the model in explaining the experiment. Lack of fit can be used to analyse parameters of the experiments to determine the suitability of the model. It is usually recommended to be higher than 0.1.

## 3. Results

The regression coefficients and analysis of variance (ANOVA) of the response as a function of the independent variables on vacuum-fried silver herring are presented in Appendix A. In addition, the experimental and predicted results of the central composite rotatable design for VF silver herring are shown in Appendix A, respectively. VF uses lower temperature than traditional frying to remove moisture from the food and reduce the oil content in the final product. The results indicated that VF temperatures had significant impacts on Aw, moisture content, yield, oil content, lightness (*L** value), and colour difference (*ÄE*). Higher VF temperatures resulted in lower Aw, moisture content, yield, and oil content of the product, but higher *L** value and Δ*E*. Next, the longer the VF duration, the higher oil content of the product. Maltose concentration was significantly and positively correlated with the yield and fracturability of the product.

### 3.1. Changes in Aw

Following a statistical regression analysis of the Aw data, the prediction model of the established quadratic polynomial regression equation was as follows:*Y* = 0.16 − 0.12X_1_ − 0.051X_2_ + 0.024X_3_ + 0.057X_1_X_2_ − (3.250E − 003)X_1_X_3_ − (6.833E − 0003)X_2_X_3_ + 0.092X_1_^2^ + 0.077X_2_^2^ − 0.040X_3_^2^(3)
where *Y* is the Aw, X_1_ is the VF temperature (°C), X_2_ is the VF duration (min), and X_3_ is the maltose concentration (%). Regression coefficients and analysis of variance (ANOVA) of the Aw as a function of the independent variables on vacuum fried silver herring are shown in Appendix A.

Based on the BBD variance analysis (Appendix A), the *p* value of this model was 0.0428 < 0.05, which meant that under the interactive effects of the three factors, the model of the quadratic polynomial regression equation had significant impacts on Aw. For the one-order analysis, the *F* value of VF temperature (X_1_) was the highest at 25.21, indicating that the impact of VF temperature (X_1_) on Aw was higher than that of VF duration (X_2_) and maltose concentration (X_3_). The impact was also significant (*p* < 0.05). *R*^2^ was 0.9028, indicating that the prediction model could explain 90.28% of the total variance. The *p* value of 0.0526 for the lack-of-fit test was not significant (*p* > 0.05), indicating that the established regression equation actually reflected the relationship between the factors and Aw (*Y*).

For the one-order analysis, VF temperature (X_1_) and duration (X_2_) had relatively large impacts on Aw. Hence, the Aw response surface and contour plots were prepared after these two factors were optimized (Figure 1). When the maltose concentration was fixed at 15%, the RSM analysis showed that Aw decreased with increasing VF temperature (X_1_) and duration (X_2_). The contour plot indicated that VF must be carried out at a high temperature or for a long time to reach the lowest Aw. In this experiment, the lowest Aw was obtained when the VF temperature and duration were 105 °C and 45 min, respectively. Our findings were similar to those of Reis et al., (2008) [24], namely, the higher the VF temperature and the longer the VF duration, the lower the Aw of the fried food product. The interaction between the variable factors is significant if the contour plot presents an elliptical shape, but not significant if the plot is circular [25,26]. The interaction between the two factors of VF temperature (X_1_) and duration (X_2_) was significant because the Aw contour plot of our results tended to be elliptical.

### 3.2. Changes in Moisture Content

The prediction model of the established quadratic polynomial regression equation following a statistical regression analysis of the data on moisture content was as follows:*Y* = 2.67 − 0.9X_1_ − 0.34X_2_ + 0.35X_3_ − 0.42X_1_X_2_ + 0.12X_1_X_3_ − 0.28X_2_X_3_ + 0.44X_1_^2^ + 0.25X_2_^2^ + 0.14X_3_^2^(4)
where *Y* is moisture content, X_1_ is VF temperature (°C), X_2_ is VF duration (min), and X_3_ is maltose concentration (%). Regression coefficients and analysis of variance (ANOVA) of the moisture content as a function of the independent variables on vacuum fried silver herring are shown in Appendix A.

Based on the BBD variance analysis (Appendix A), the *p* value of this model was 0.0917 > 0.05, indicating that the quadratic polynomial regression equation had no significant impact on the moisture content following the interactions of these three factors. The *F* value of 17.63 for VF temperature (X_1_) was the highest in the one-order analysis, indicating that its impact on moisture content was higher than that of VF duration (X_2_) and maltose concentration (X_3_). Furthermore, the impact was significant (*p* < 0.05). *R*^2^ was 0.9034, meaning that the prediction model could explain 90.34% of the total variance. The value of *p* in the lack-of-fit test was 0.1554, which was not significant (*p* > 0.1). This indicated that the established regression equation actually reflected the relationship between the factors and moisture content (*Y*).

During the one-order analysis, VF temperature (X_1_) and maltose concentration (X_3_) had relatively large impacts on moisture content. Hence, the response surface and contour plots for moisture content were prepared after these two factors were optimized (Figure 2). When the VF duration (X_2_) was fixed at 35 min, the RSM analysis showed that the higher the VF temperature (X_1_), the lower the moisture content. The moisture content was the lowest when the VF temperature (X_1_) was the highest (105 °C). The contour plot tended to be circular, indicating that the interaction between the two factors of VF temperature and maltose concentration was not significant. The impact of maltose concentration on moisture content was also not significant.

### 3.3. Changes in Yield

Following a statistical regression analysis of the yield data, the prediction model of the established quadratic polynomial regression equation was as follows:*Y* = 22.96 − 2.09X_1_ + 7.500E − 003X_2_ + 3.53X_3_ + 0.64X_1_X_2_ − 0.33X_1_X_3_ + 0.79X_2_X_3_ + 0.35X_1_^2^ + 0.79X_2_^2^ + 0.33X_3_^2^(5)
where *Y* is the yield, X_1_ is the VF temperature (°C), X_2_ is the VF duration (min), and X_3_ is the maltose concentration (%). Regression coefficients and analysis of variance (ANOVA) of the yield as a function of the independent variables on vacuum fried silver herring are shown in Appendix A.

Based on the BBD variance analysis (Appendix A), the *p* value of this model was 0.0575 > 0.05, indicating that the impact of the quadratic polynomial regression equation on the yield was not significant following the interaction of these three factors. For the one-order analysis, the *F* value of 28.07 for maltose concentration (X_3_) was the highest, showing that its impact on the yield was very significant (*p* < 0.01) and higher than that of VF temperature (X_1_) and duration (X_2_). The VF temperature (X_1_) also had a significant impact on the yield (0.05 > *p* > 0.01). However, the *p* value of the VF duration (X_2_) was 0.9914, showing that its impact on the yield was minimal. *R*^2^ was 0.8890, indicating that the prediction model could explain 88.90% of the total variance. The value of *p* in the lack-of-fit test was 0.0585, which was not significant (0.1 > *p* > 0.05). This indicated that the established regression equation accurately reflected the relationship between the factors and the yield (*Y*).

The one-order analysis revealed that VF temperature (X_1_) and maltose concentration (X_3_) had relatively large impacts on the yield. Therefore, the response surface and contour plots of the yield were prepared after these two factors were optimized (Figure 3). When the VF duration (X_2_) was fixed at 35 min, the RSM analysis showed that the yield increased with increasing maltose concentrations (X_3_). However, the higher the VF temperature (X_1_) resulted in the lower the yield. When the VF temperature was 105 °C, the yield of the sample was lower than 20%. Because the purpose of this study was not to improve the yield, the results were not listed in the follow-up optimization conditions.

### 3.4. Changes in Oil Content

After a statistical regression analysis of the oil content data, the prediction model established by the quadratic polynomial regression equation was as follows:*Y* = 24.95 − 1.1X_1_ + 1.47X_2_ − 0.88X_3_ + 1.06X_1_X_2_ − 1.15X_1_X_3_ − 0.19X_2_X_3_ + 1.17X_1_^2^ + 1.59X_2_^2^ − 1.4X_3_^2^(6)
where *Y* is oil content, X_1_ is VF temperature (°C), X_2_ is VF duration (min), and X_3_ is maltose concentration (%). Regression coefficients and analysis of variance (ANOVA) of the oil content as a function of the independent variables on vacuum fried silver herring are shown in Appendix A.

Based on the BBD variance analysis (Appendix A), the *p* value of this model was 0.0319 < 0.05, indicating that the quadratic polynomial regression equation had a significant impact on oil content following the interaction of these three factors. The *F* value for VF duration (X_2_) was the highest at 14.02, indicating that the impact on oil content was higher than that of VF temperature (X_1_) and maltose concentration (X_3_). The impact was also significant (*p* < 0.05). The coefficient of determination *R*^2^ was 0.9146, meaning that the prediction model could explain 91.46% of the total variance. The *p* value of 0.7440 for the lack-of-fit model was not significant (*p* > 0.1). This showed that the established regression equation reflected the relationship between the factor and oil content (*Y*). In summary, the regression model was reasonable and the equation could be used to predict oil content.

Under the one-order analysis, the impacts of VF temperature (X_1_) and duration (X_2_) on oil content were relatively large. Therefore, after these two factor variables were optimized, the response surface and contour plots of oil content were drawn (Figure 4). When the maltose concentration (X_3_) was fixed at 15%, the RSM analysis results showed that the higher the VF temperature had the lower the oil content of the sample. In contrast, the longer the VF duration resulted in the higher the oil content. In the contour plot, the relationship between VF temperature and duration was close to an ellipse, indicating that the two factors had significant impacts on oil content. Samples with the lowest oil content were produced when the VF temperature was between 99 °C and 104 °C and the VF duration was between 26 and 28 min.

### 3.5. Changes in the L* Value

The prediction model of the established quadratic polynomial regression equation, following the statistical regression analysis of the *L** value data, was as follows:*Y* = 26.99 + 1.73X_1_ − 0.3X_2_ − 0.5X_3_ − 0.91X_1_X_2_ − 0.42X_1_X_3_ + 0.36X_2_X_3_ − 0.062X_1_^2^ + 0.39X_2_^2^ − 1.11X_3_^2^(7)
where *Y* is the *L** value, X_1_ is the VF temperature (°C), X_2_ is VF duration (min), and X_3_ is the maltose concentration (%). Regression coefficients and analysis of variance (ANOVA) of the *L** as a function of the independent variables on vacuum fried silver herring are shown in Appendix A.

Based on the BBD variance analysis (Appendix A), the *p* value of this model was 0.0554 > 0.05, indicating that the quadratic polynomial regression equation did not have any significant impact on the *L** value after the interaction of these three factors. However, in the one-order analysis, the VF temperature (X_1_) had the highest F value of 26.68. This indicated that the impact of X_1_ on the *L** value was higher than that of VF duration (X_2_) and maltose concentration (X_3_), and that the impact was very significant (*p* < 0.01). The *R*^2^ value of 0.8908 indicated that the prediction model could explain 89.08% of the total variance. The *p* value of 0.8873 for the lack-of-fit test was not significant (*p* > 0.1), indicating that the established regression equation could reflect the relationship between the factors and the *L** value. Based on the above, the regression model was reasonable and the equation could be used to predict the *L** value.

For the one-order analysis, VF temperature (X_1_) and maltose concentration (X_3_) had relatively large impacts on the *L** value. Therefore, after optimizing these two factor variables, the response surface and contour plots of the *L** value were prepared (Figure 5). When the VF duration (X_2_) was fixed at 35 min, the RSM analysis showed that as the VF temperature (X_1_) increased, the *L** values of the samples also increased significantly. Although increasing maltose concentrations (X_3_) caused the *L** value to rise, the *L** inversed and decreased when the concentration exceeded 16%. The surface of the samples became darker, which might be related to the browning caused by the Maillard reaction of excess maltose and amino acids. The contour plot shows that the relationship between VF temperature (X_1_) and maltose concentration (X_3_) tended to be elliptical, which meant that the two factors had significant impacts on the *L** value.

### 3.6. Changes in ÄE

Following a statistical regression analysis of the *ÄE* data, the prediction model of the quadratic polynomial regression equation was as follows:*Y* = 27.2 + 1.72X_1_ − 0.3X_2_ − 0.53X_3_ − 0.92X_1_X_2_ − 0.41X_1_X_3_ + 0.36X_2_X_3_ − 0.094X_1_^2^ + 0.4X_2_^2^ − 1.1X_3_^2^(8)
where *Y* is *ÄE*, X_1_ is VF temperature (°C), X_2_ is VF duration (min), and X_3_ is maltose concentration (%). Regression coefficients and analysis of variance (ANOVA) of the *ÄE* as a function of the independent variables on vacuum fried silver herring are shown in Appendix A.

Based on the BBD variance analysis (Appendix A), the values of *ÄE* and *L** had the same trend under the BBD variable analysis. The *p* value of this *ÄE* model was 0.0576 > 0.05, indicating that the impact of the quadratic polynomial regression equation model was not significant. The *R*^2^ was 0.8888, which meant that the prediction model could explain 88.88% of the total variance. The value of *p* in the lack-of-fit test was 0.9056. This was not significant (*p* > 0.1), indicating that the established regression equation reflected the relationship between the factors and *ÄE*.

For the one-order analysis, the *F* value of VF temperature (X_1_) was the highest at 25.81 and the impact was significant (*p* < 0.01). This *F* value was much higher than that of VF duration (X_2_) and maltose concentration (X_3_), indicating that X_1_ had a significant impact on *ÄE* and that the impact was much higher than that of X_2_ and X_3_. The regression model was deemed to be reasonable based on the above, and the equation could be used to predict *ÄE*.

The impacts of VF temperature (X_1_) and maltose concentration (X_3_) on the *ÄE* of silver herring were relatively large during the one-order analysis. Hence, the response surface and contour plots of *ÄE* were drawn after these two factor variables were optimized (Figure 6). This figure presents similar results to that for the *L** values (Figure 5). In other words, when the VF duration (X_2_) was fixed at 35 min, the hotter the VF temperature (X_1_) resulted in the higher the *ÄE*. Although the *ÄE* increased with an increasing maltose concentration (X_3_), the *ÄE* inversed and decreased when the concentration exceeded 16%. This meant that the colour difference on the surface of the samples diminished and they tended to become darker. This might be related to the browning caused by the Maillard reaction arising from excess maltose and amino acids. The contour plot showed that the relationship between VF temperature (X_1_) and maltose concentration (X_3_) tended to be elliptical, indicating that both factors had significant impacts on the *ÄE* value.

### 3.7. Changes in Fracturability

After the statistical regression analysis of the fracturability data, the prediction model of the quadratic polynomial regression equation was established as follows:*Y* = 284.69 − 16.13X_1_ + 27.27X_2_ + 76.29X_3_ + 5.79X_1_X_2_ + 74.32X_1_X_3_ + 9.37X_2_X_3_ + 36.2X_1_^2^ + 29.27X_2_^2^ + 41.85X_3_^2^(9)
where *Y* is fracturability, X_1_ is VF temperature (°C), X_2_ is VF duration (min), and X_3_ is maltose concentration (%). Regression coefficients and analysis of variance (ANOVA) of the fracturability as a function of the independent variables on vacuum fried silver herring are shown in Appendix A.

Based on the BBD variance analysis (Appendix A), the *p* value of this model was 0.1300 > 0.05, which meant that the quadratic polynomial regression equation had no significant impact on fracturability following the interaction of these three factors. Nevertheless, the prediction model could explain 83.73% of the total variance because the value of the coefficient of determination *R*^2^ was 0.8373. The *p* value of the lack-of-fit model was 0.0524, which was not significant (*p* > 0.05). This indicated that the established regression equation truly reflected the relationship between the factors and fracturability. The *F* value of maltose concentration (X_3_) during the one-order analysis was the highest at 13.34 and the impact was significant (0.01 < *p* < 0.05). This *F* value was much higher than those for VF temperature (X_1_) and duration (X_2_), which were 0.60 and 1.71, respectively. This indicated that the impact of maltose concentration on fracturability was higher than those of VF temperature and duration.

Under the one-order analysis, VF duration (X_2_) and maltose concentration (X_3_) had relatively large impacts on the fracturability of the fried silver herring. After those two factors were optimized, the response surface and contour plots were prepared (Figure 7). When the VF temperature (X_1_) was fixed at 90 °C, the RSM analysis showed that the higher the maltose concentration had the higher the fracturability. The products had less fracturability when the maltose concentration was between 0 and 6%. Next, the longer the VF duration resulted in the lower the fracturability. When the duration was between 28 and 35 min, fracturability was relatively poor. The contour plot showed that the relationship between the VF temperature (X_1_) and maltose concentration (X_3_) was close to an ellipse, indicating that the two factors had significant impacts on fracturability. Overall, the findings showed a strong relationship between maltose concentration and fracturability.

### 3.8. Analysis of Optimal VF Conditions for Silver Herring

The Design-Expert 7.0 (Stat-Ease Inc., Minneapolis, MN, USA) statistical software was used to analyse the results of the various combinations of samples and optimal conditions. The optimal regression model was arrived at after preparing a response surface plot (Figure 8). Specifically, the optimal combination of processing conditions was a VF temperature (X_1_) of 105 °C, VF duration (X_2_) of 25 min, and maltose concentration (X_3_) of 27%. The resultant quality values of the fried silver herring products were as follows: moisture content at 3.91%, Aw at 0.198, oil content at 21.69%, *L** value at 28.19, *ÄE* at 27.31, and fracturability at 359 g/s.

## 4. Discussion

According to the results of this study, VF temperature had a significant impact on Aw and moisture content: the higher the VF temperature, the lower the Aw and moisture content of the products. These results were similar to those of Pan et al. [19] and Andres-Bello et al. [20], who stated that the moisture content of VF breaded shrimps and sea bream fillets decreased with rising VF temperatures. Also, the oil content reduced with increasing temperatures. Andres-Bello et al. [20] reported similar results for VF of sea bream fillets: the oil absorption of the samples reduced when the VF temperature increased (90–110 °C), and fish fillets prepared at the highest temperature of 110 °C had the lowest oil content. In contrast, the longer the VF duration, the higher the oil content tended to be. The results of this study were also similar to the reports of Pan et al. [19] and Andres-Bello et al. [20], who concluded that the oil content of VF breaded shrimps and sea bream fillets increased with longer VF durations. Most studies on VF fruit, vegetable, and potato chips had similar findings to ours [10,12,24,25,26]. Akinpelu et al. [12] proposed that the oil absorption amount was positively correlated with VF duration in a VF system.

In terms of yield, the higher the VF temperature resulted in the lower the yield of the samples. Andres-Bello et al. [20] similarly reported that the weight loss of VF sea bream fillets increased with higher VF temperatures. This study further found a significantly positive correlation between maltose concentration and yield: the stronger the concentration, the higher the yield. Shyu et al. [10] immersed carrot chips in fructose before VF and found that the longer the duration of immersion, the higher the sugar content in the products. When fried in this state, the products retained more moisture but had a lower oil content. Consequently, the weight loss of the samples was reduced. Jayaraman et al. [27] proposed that immersing cauliflower in a salt and sucrose solution before dehydration could reduce shrinkage and preserve the integrity of the cell walls. It was deduced that the maltose molecules entered the tissues of the ingredients due to osmotic pressure. This led to the retention of additional water molecules and reduced oil absorption during subsequent frying. As a result, the appearance and cell integrity of the products were maintained, weight loss was reduced, and the yield increased.

In terms of the colour value, this study found that the higher the VF temperature had the higher the *L** value of the samples. This result was contrary to the findings of previous research on VF vegetable, fruit, and potato chips; sea bream fillets; and breaded shrimps, most of which reported that the higher the VF temperature had the lower the *L** value and the darker the appearance [10,12,19,20,24,25]. The main cause for this difference was the skin of the silver herring, which is covered in small, silvery-white scales. The skin became brighter and whiter when heated at high temperatures, resulting in an increase in the *L** value [1]. Next, the higher the VF temperature resulted in the higher the *ÄE* of the samples, which was similar to the results of previous research [10,12,19,20,24,25]. The main reason was that high-temperature thermal treatment increased the rate of non-enzymatic browning reaction and products [10,19,20].

In general, dehydration caused by the frying process leads to textural changes in the samples, such as increased crispness of the product. In this study, fracturability was used as an indicator of the crispness of the fried sliver herring. Specifically, the higher the fracturability had the crisper the product. This study found that the stronger the maltose concentration, the higher the fracturability. The possible reason was that malt sugar molecules entered the tissues of the ingredients due to osmotic pressure, whereas water molecules flowed out, leading to dehydration. The result was greater crispness [4] after VF.

## 5. Conclusions

After analysing the results of fried silver herring using RSM, we found that VF temperature, VF duration, and maltose concentration all affect the main quality characteristics of fried silver herring, especially in terms of oil content. The optimal combination of conditions was found to be a VF temperature of 105 °C, VF duration of 25 min, and maltose concentration of 27%. This was the first study in which the maltose concentration for immersion of the ingredients during pre-processing was included as one of the parameters for VF of fish and combined with other factors to examine the impacts on product quality. Therefore, vacuum frying is a viable option applied to produce fried silver herring products with better quality.

## Figures and Tables

**Figure 1 foods-12-03533-f001:**
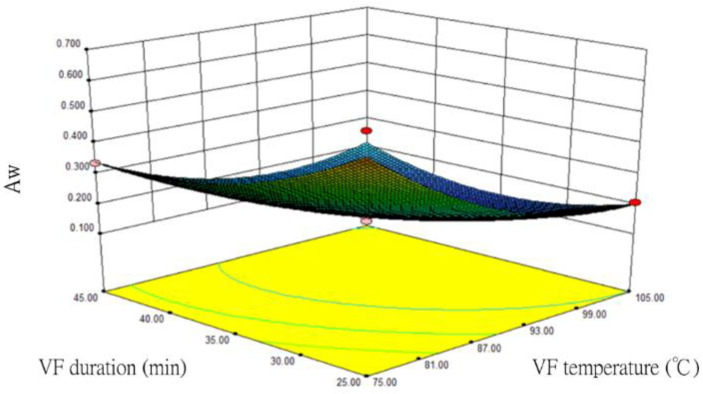
The response surface and contour plots for the impacts of vacuum frying (VF) temperature and duration on the Aw in fried silver herring (maltose concentration at 15%).

**Figure 2 foods-12-03533-f002:**
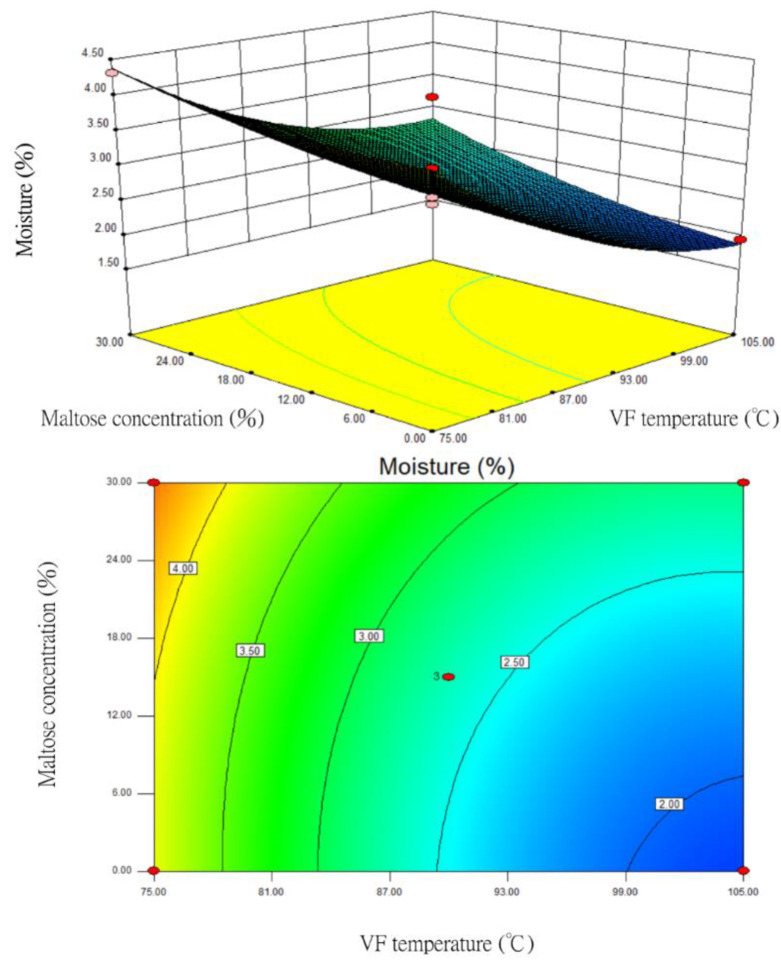
The response surface and contour plots for the impacts of VF temperature and maltose concentration on the moisture content in fried silver herring (VF duration of 35 min).

**Figure 3 foods-12-03533-f003:**
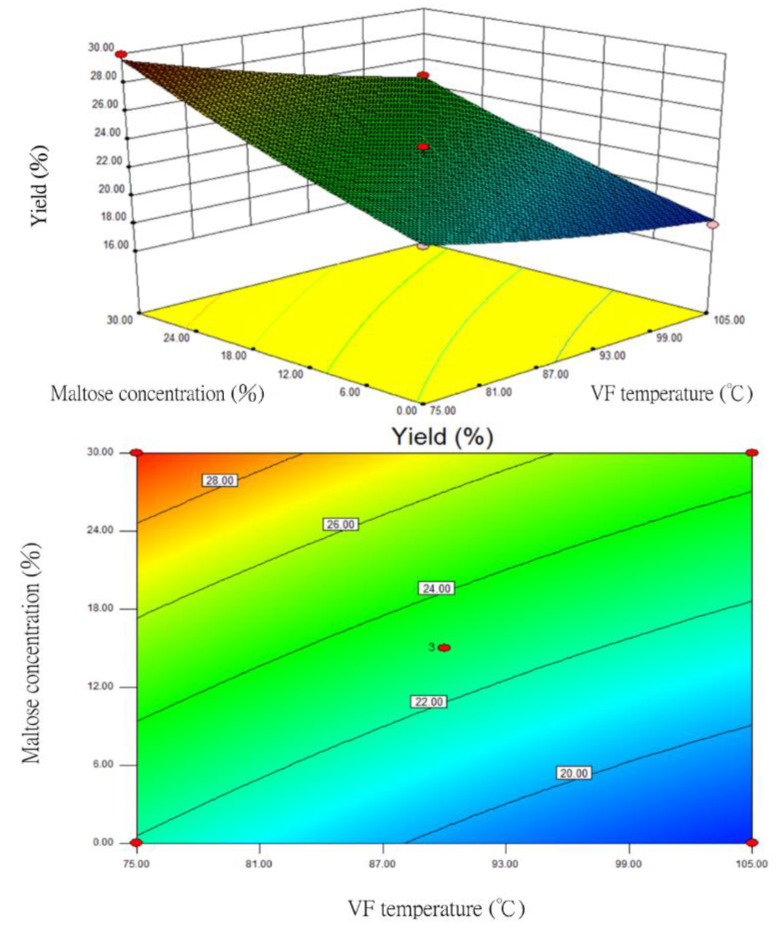
The response surface and contour plots for the impacts of VF temperature and maltose concentration on the yield of fried silver herring (VF duration of 35 min).

**Figure 4 foods-12-03533-f004:**
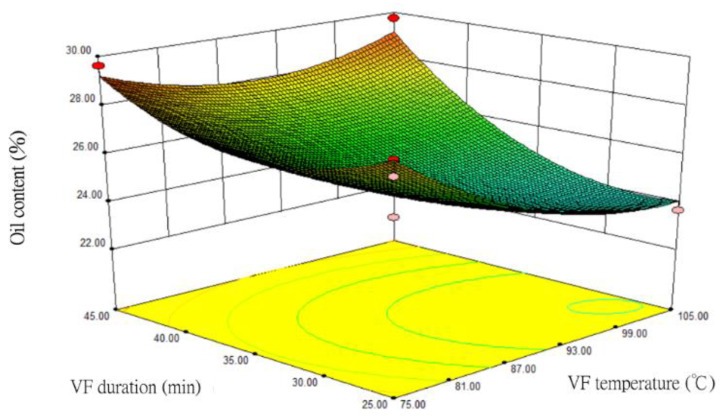
The response surface and contour plots for the impacts of VF temperature and duration on the oil content of fried silver herring (maltose concentration at 15%).

**Figure 5 foods-12-03533-f005:**
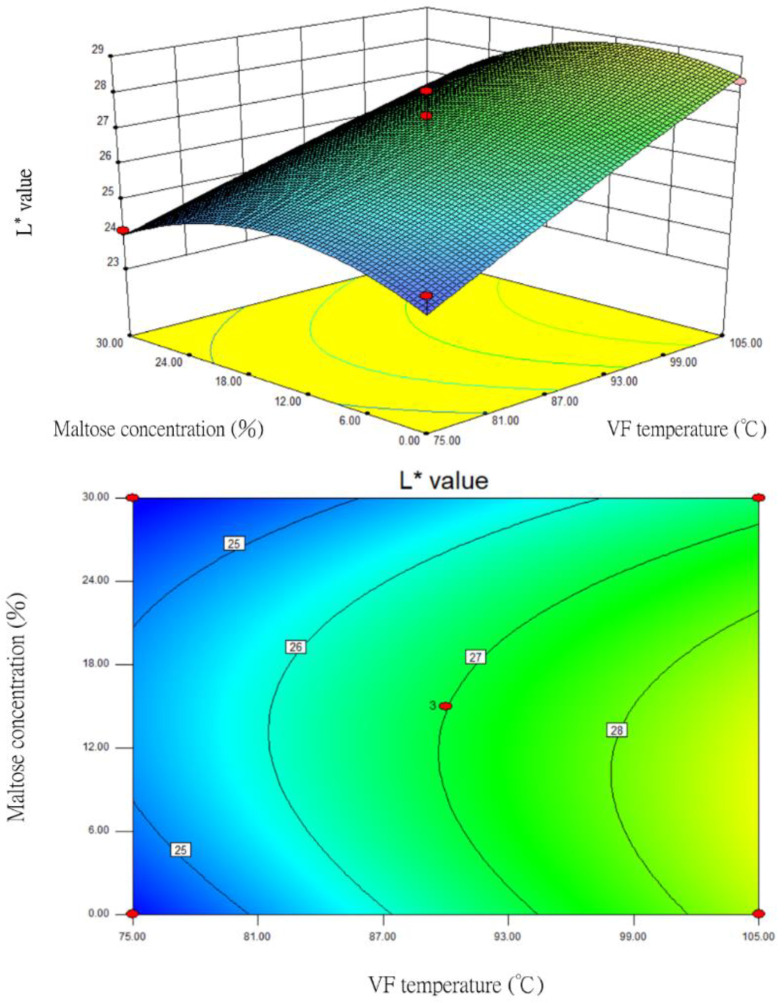
The response surface and contour plots for the impacts of VF temperature and maltose concentration on the *L** value of fried silver herring (VF duration of 35 min).

**Figure 6 foods-12-03533-f006:**
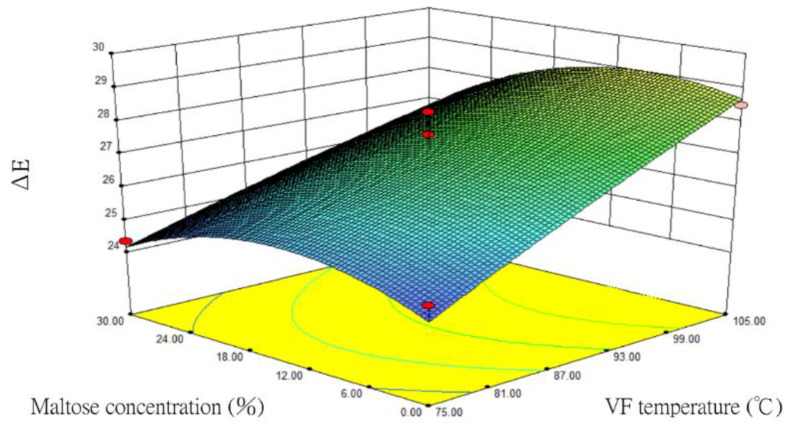
The response surface and contour plots for the impacts of VF temperature and maltose concentration on the *ÄE* value of fried silver herring (VF duration of 35 min).

**Figure 7 foods-12-03533-f007:**
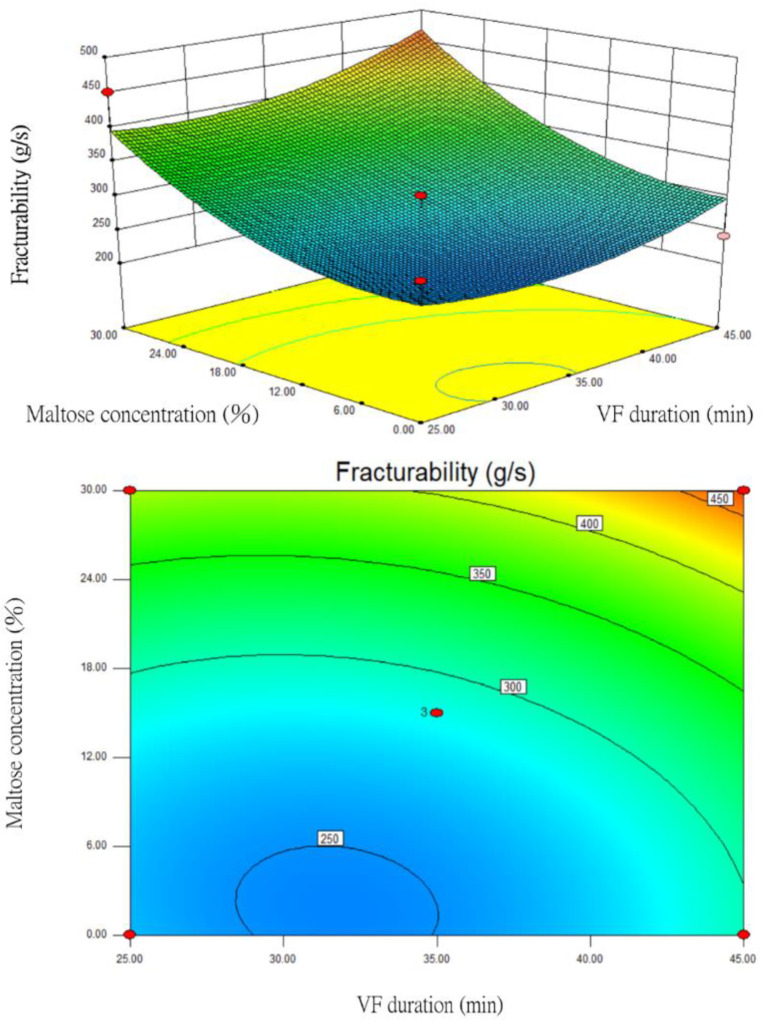
The response surface and contour plots for the impacts of VF duration and maltose concentration on the fracturability of fried silver herring (VF temperature at 90 °C).

**Figure 8 foods-12-03533-f008:**
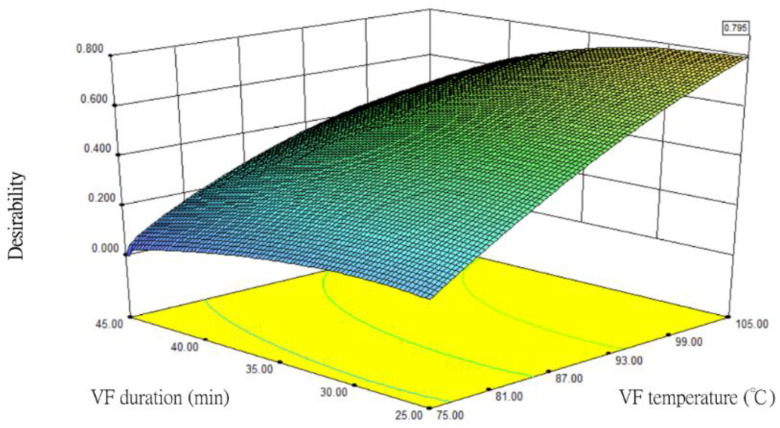
The response surface plot for the optimal VF conditions for fried silver herring. VF temperature of 105 °C, VF duration of 25 min, and maltose concentration of 27%.

**Table 1 foods-12-03533-t001:** The levels and ratios of the response surface methodology design with different operation conditions.

Treatment	X_1_	X_2_	X_3_
1	75 (−1)	25 (−1)	15 (0)
2	105 (+1)	25 (−1)	15 (0)
3	75 (−1)	45 (+1)	15 (0)
4	105 (+1)	45 (+1)	15 (0)
5	75 (−1)	35 (0)	0 (−1)
6	75 (−1)	35 (0)	30 (+1)
7	105 (+1)	35 (0)	0 (−1)
8	105 (+1)	35 (0)	30 (+1)
9	90 (0)	25 (−1)	0 (−1)
10	90 (0)	25 (−1)	30 (+1)
11	90 (0)	45 (+1)	0 (−1)
12	90 (0)	45 (+1)	30 (+1)
13 (Centre point)	90 (0)	35 (0)	15 (0)
14 (Centre point)	90 (0)	35 (0)	15 (0)
15 (Centre point)	90 (0)	35 (0)	15 (0)

X_1_ = VF temperature (°C), X_2_ = VF duration (min), X_3_ = Maltose concentration (%).

## Data Availability

The data presented in this study are available on request from the corresponding author. The data are not publicly available due to privacy and ethical reasons.

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
