# Peer review of "Determining the Optimal Vacuum Frying Conditions for Silver Herring (Spratelloides gracilis) Using the Response Surface Methodology"

_foods, 2023, doi:10.3390/foods12193533_

Round 1

Reviewer 1 Report (Previous Reviewer 1)

The authors have attended to all the reviewers' comments satisfactorily.

Reviewer 2 Report (Previous Reviewer 2)

The manuscript was revised previously by this reviewer. The authors have significantly improved the MS and responded satisfactorily to all comments and suggestions made by this reviewer. In my opinion, no further revisions are required.

This manuscript is a resubmission of an earlier submission. The following is a list of the peer review reports and author responses from that submission.

Round 1

Reviewer 1 Report

L21: Oil content or oil uptake or oil absorption. Also, please check if it should be ΔE or ÄE.

L39-40: A scientific explanation of this traditional process would be helpful. L91-94: How were these factors and levels selected? Is it from any previous studies or based on some other preliminary results? L103: A few references for using this processing condition are required. Is there any specific fish-to-water ratio used for boiling? If so, then it should be mentioned with proper citations. L108-112: Reason for selecting these specific processing parameters. Are these due to any instrumental limitations or based on past literature? L109: Any specific explanation for selecting the BBD model? L119: How did the author decide upon these specific processing parameters?
L125-126: This equation is in image format and needs to be depixelated for better readability. The author may use any specific tool (e.g., MathType) to write mathematical or chemical equations. L164-167: Are these parameters based on any standards protocol in the instrument or based on past literature?   Major comments 1. An explanation for the physical significance of the magnitude and sign of the coefficients obtained in the equations (in L188-189, L224-225, L255-256, L288-289, L321-322, L357-358, L394-395)  is required.  2. There are a few cases where the p-value of the equation is not significant (L228, L259, L325, L362, L398). The author may fit the result using equations other than the quadratic one. 3. What was the relative importance of each response taken into consideration? If so, then an explanation is required. 4. How is this optimized value obtained in L427-434? What kind of optimization (numerical or graphical) was used in this study? Also, What goals were set for each factor while optimization? 5. What is the meaning of "product quality" mentioned in Fig. 8 (L430)? Is it the "overall product quality" which is a function of other responses? If so, then an explanation is required.   General comments The equations used could have been numbered for better reading and reviewing purposes. What was the control sample used in this study? Any comparative analysis with traditional frying?
References 22 and 23, mentioned at the end, are NOT cited in the paper.

-Poor English language, grammar, and formatting used.

Reviewer 2 Report

The authors determined the effect and interactions of temperature, time, and maltose concentration, as processing conditions during the vacuum frying process of silver herring, on the quality of the finished product. Also, RSM was used to establish optimal processing conditions. The topic of the study is novel, interesting, and fits perfectly into the Foods journal scope. However, there are many inconsistencies in the materials and methods and results sections. The authors should have provided more information about the sample number analyzed in each treatment combination in the materials and method section. Therefore, it is impossible to determine if the presented results and conclusions were driven by a representative data population. Also, the authors did not provide any information about the optimization process (technique used, target conditions during the optimization) and validation of the model (optimal processing conditions should be proved experimentally and statistically analyzed). On the other hand, the results are not discussed appropriately; the authors should explain how the main factors and interactions affected all response variables. Also, important statistical information about the models needs to be included in the text (a table summarizing statistical information is suggested).

Substantial changes are needed for this article.

Concerning the text, some points of clarification and suggestions are indicated below.

L17. Change 105°C to 105 °C (in a separate form). Check all that apply in the manuscript.

L17. Change mins to min. Check all that apply in the manuscript.

L21. Change ÄE to ΔE. Check all that apply in the manuscript.

L29. Include a general conclusion.

L30-31. Include keywords different from the ones used in the title.

L103. Change seconds to s. Check all that apply in the manuscript.

L111. Explain how the factor levels were selected.

L113. Include the number of experimental units used in each treatment combination.

L119. Change hour to h. Check all that apply in the manuscript.

L122. The factors for the sample testing need to be corrected. Measurements refer to response variables.

L122-123. Include units in each measurement.

L123-129. This paragraph is part of the statistical analysis.

L130. Please include how many replicates per combination of treatments were used. Include this information in all response variables analyzed.

L131. Change AW to Aw

L132. Specify conditions before Aw measurement (number of samples used, sample grinding, etc.).

L164. In fracturability measurement, include units.

L167-168. Include a section related to the optimization and specify which technique and how the optimal conditions were carried out. Specify what were the target conditions during optimization. Also, include how the validation process was carried out (optimal condition predicted by the model should be validated in the laboratory).

L173. This reviewer suggests including a table with the predicted and experimental values ​​for each response variable (considering the information in Table 1).

L173. The models of all the response variables described in the text can be summarized in a table. This table could also include statistical data for each model (significance of model terms, F-value, p-value, R2, and lack of fit).

L184. Include the lack of fit of the model.

L174-184. This paragraph corresponds to the methodology.

L185. A more detailed explanation should be included about the effect of the main factors and interactions on all response variables studied (Aw, moisture content, yield, oil content, L*-value, ΔE, fracturability).

L188-189. How do the authors explain the main effects? How do the authors explain the interaction effects? How do the authors explain the quadratic effects? These questions apply to all the response variables studied.

L215-217. Improve graph quality. This recommendation applies to all graphs.

L391. Include a number section.

L426. Only models with p-value < 0.05, R2 near 1, and lack of fit p-value > 0.05 should be included in the analysis of optimal conditions.

L438-488. This paragraph is part of a discussion about the effects of the factors on these response variables and is not part of the optimization process.

L489. A conclusion is not a summary of the results; please rewrite the conclusion.

L518. Update some references in the manuscript.

Reviewer 3 Report

Dear Authors, please find below detailed comments on the manuscript:

1) Both in Abstarct and in Introduction (the final part) the purpose of the work should be clearly defined.

2) The formulas should be corrected in accordance with the editorial requirements (the template number in the text - as a reference, the template number consecutively),

3) Statistical analysis - what was the probability level (p)?

4) It is necessary to improve the readability of drawings (axle descriptions)

5) 4. Conclusion - the chapter is a summary, I suggest changing it to "Summary"

6) The literature on vacuum frying is very extensive, please expand with the latest bibliographic items
